# Precise Simulation of Heat-Flow Coupling of Pipe Cooling in Mass Concrete

**DOI:** 10.3390/ma14185142

**Published:** 2021-09-08

**Authors:** Peng Yu, Ruiqing Li, Dapeng Bie, Xiancai Liu, Xiaomin Yao, Yahui Duan

**Affiliations:** 1Hubei Institute of Water Resources Survey and Design, Wuhan 430070, China; yupeng@whu.edu.cn (P.Y.); linpr0626@gmail.com (X.L.); hydroyw@126.com (X.Y.); 2State Key Laboratory of Water Resources and Hydropower Engineering Science, Wuhan University, Wuhan 430072, China; duanyahui@whu.edu.cn; 3School of Urban Construction, Wuchang University of Technology, Wuhan 430223, China

**Keywords:** mass concrete, pipe cooling, heat-flow coupling, numerical simulation

## Abstract

For a long time, temperature control and crack prevention of mass concrete is a difficult job in engineering. For temperature control and crack prevention, the most effective and common-used method is to embed cooling pipe in mass concrete. At present, there still exists some challenges in the precise simulation of pipe cooling in mass concrete, which is a complex heat-flow coupling problem. Numerical simulation is faced with the problem of over-simplification and inaccuracy. In this study, precise simulation of heat-flow coupling of pipe cooling in mass concrete is carried out based on finite element software COMSOL Multiphysics 5.4. Simulation results are comprehensively verified with results from theoretical solutions and equivalent algorithms, which prove the correctness and feasibility of precise simulation. Compared with an equivalent algorithm, precise simulation of pipe cooling in mass concrete can characterize the sharp temperature gradient around cooling pipe and the temperature rise of cooling water along pipeline more realistically. In addition, the cooling effects and local temperature gradient under different water flow (0.60 m^3^/h, 1.20 m^3^/h, and 1.80 m^3^/h) and water temperature (5 °C, 10 °C, and 15 °C) are comprehensively studied and related engineering suggestions are given.

## 1. Introduction

During the casting of mass concrete, the hydration of cement releases a lot of heat, and the internal temperature of concrete rises immediately, followed by a temperature drop under the effect of boundary heat dissipation. Under the restraint of foundation and concrete, the temperature variation and gradient in concrete are easy to cause cracking [1]. In order to prevent temperature cracks, the U.S. Bureau of Reclamation first proposed to embed cooling pipe in concrete for temperature control, which has achieved good applications in engineering practices [2,3,4,5].

As for the research of pipe cooling, many scholars have carried out field experiments, where traditional thermometer or optical-fiber temperature measurement was adopted to monitor the temperature gradient around cooling pipe in mass concrete [6,7,8,9]. It was pointed out that the water flow and water temperature directly affect the cooling efficiency and the temperature gradient around the cooling pipe [8,9].

It is rather complicated to precisely calculate the thermal field of concrete with the cooling pipe embedded. On one hand, compared with concrete structure, the size of cooling pipe is relatively small, which makes the grid generation difficult; On the other hand, the thermal field of concrete containing cooling pipe is a typical non-stationary heat-flow coupling field with varying boundary temperature: the cooling effect relates to not only water temperature, but also water flow.

Over the past thirty years, scholars have carried out a lot of research on calculation methods for the thermal field of concrete with cooling pipe embedded. In engineering, the most widely used method is the equivalent approach [10,11,12,13]. In the equivalent approach, pipe cooling is equivalent to a “negative heat source” to reduce heat releasing during hydration [10,11]. This method does not need to model cooling pipes and has the advantages of simple pre-treatment and high calculation efficiency. Regrettably, the equivalent approach cannot describe the sharp temperature gradient around the cooling pipe and the temperature rise of water flow along the pipeline. While, many engineering examples show that a sharp cooling may produce a large global temperature gradient and tensile stress, which may lead to the generation and expansion of cracks from inside [14,15,16,17].

In terms of precise numerical simulation of pipe cooling, Kim et al. [18] implemented a line element for the precise modeling of cooling pipe and calculated the thermal field of concrete with pipe cooling using internal flow theory. Chen et al. [19,20,21] developed a composite element method (CEM) and took each pipe segment as a sub-element embedded in concrete elements. Zhu et al. [22] proposed a three-dimensional calculation method of pipe cooling, which strictly simulated the actual layout of the cooling pipe and calculated the increment of water temperature along the pipeline. Liu [23] proposed a variety of division methods of cooling pipe surrounding units using substructure technology. Zhou et al. [24,25] established a heat-flow coupling model in finite element software Ansys via secondary development technology. Hong et al. [26] simulated of thermal field in a mass concrete structure with the cooling pipe by the localized radial basis function collocation method.

Although many scholars have done pioneering research, there still exist some issues that should be further studied: (1) most current simulations adopt simplified integral to approximate the heat-flow coupling on cooling pipe, which cannot truly reflect the physical boundary. (2) In numerical solution, an iterative calculation is usually needed for calculating the temperature rise of cooling water along the pipeline, which seems rather complex. (3) It is difficult to simulate the real pipeline size and layout, and many simulations are simplified to a large extent.

In order to give full play to the effect of pipe cooling in temperature control and crack prevention, a precise simulation for heat-flow coupling of pipe cooling in concrete is constructed based on finite element software COMSOL Multiphysics 5.4. Simulation results are comprehensively verified and analyzed with results from theoretical solutions and equivalent algorithms, which prove the correctness and feasibility of precise simulation.

## 2. Heat-Flow Coupling of Pipe Cooling

As shown in Figure 1, the numerical simulation for concrete containing cooling pipe includes two aspects: fluid field for cooling water in cooling pipe and thermal field in concrete and cooling water. There exists a coupling relationship between the fluid and thermal fields: the calculation of thermal field lay on the basis of the fluid field, and thermal field affects model parameters in the fluid field in turn. The whole model can be divided into a solid domain, fluid domain, and thin layer, corresponding to concrete, cooling water, and intermediate cooling pipe.

The water flow in the cooling pipe is controlled by the incompressible Navier-Stokes equation, in which flow velocity u and pressure p are variables to be solved.
(1)$$\rho \frac{\partial u}{\partial t}+\rho u\nabla u=-\nabla p+\nabla (\mu (\nabla u+{\left(\nabla u\right)}^{T})-\frac{2}{3}\mu (\nabla u)I)$$
(2)∇(ρu)=0
where ρ is the density of water; μ is the dynamic viscosity, which is function of temperature [27].

For solving the fluid field, it is necessary to artificially set the flow velocity at the inlet and outlet of the cooling pipe. Assuming that pressure *p* is 1 atm, an o-slip wall boundary is applied on the inner wall of the cooling pipe. With respect to the heat-flow coupling problem, heat transfer in a fluid domain is conducted by both heat conduction and convection, while only heat conduction exists in the solid domain and thin layer. Heat-flow coupling in fluid and solid domains can be expressed by the following equations:(3)$$\rho C_{p}\frac{\partial T}{\partial t}+\rho C_{p}u\nabla T+\nabla q=Q$$
(4)q=−k∇T
where T represents the temperature to be solved, $C_{p}$ is heat capacity, k is thermal conductivity. In Equation (3), $\rho C_{p}\frac{\partial T}{\partial t}$ is the transient term, $\rho C_{p}u\nabla T$ is the convective term, ∇q is the diffusion term and Q is the heat source. With respect to the heat conduction in the concrete and cooling pipe, the convection term is zero, and Equation (3) degenerates into the Fourier equation. For the cooling water in the cooling pipe, the convection term is not zero, and the cooling effect under different water flows is reflected by the convection term.

In addition, the initial and boundary conditions of the thermal field can be written as follows:(5)$$T_{t=0}=T_{0}(x,y,z)$$
(6)$${\left.T\right|}_{\Gamma_{1}}=T_{a}\;(\mathrm{Dirichlet}\;\mathrm{boundary}\;\mathrm{condition})$$
(7)$${\left.\lambda \frac{\partial T}{\partial \{n\}}\right|}_{\Gamma_{2}}=\beta (T-T_{b})\;(\mathrm{Neumann}\;\mathrm{boundary}\;\mathrm{condition})$$

In the formula, $T_{0}(x,y,z)$ is the initial temperature field, $T_{a}$ is the known temperature on the Dirichlet boundary. $T_{b}$ is the ambient temperature on the Neumann boundary; β is the surface heat release coefficient, $\frac{\partial T}{\partial \{n\}}$ is the outer normal direction of the Neumann boundary.

## 3. Numerical Implementation Based on COMSOL Multiphysics

In this study, the multiple physical fields coupling analysis software COMSOL Multiphysics 5.4 is adopted to conduct the precise heat-flow coupling analysis. Comsol Multiphysics is numerical simulation software based on the finite element method, which establishes a model on the basis of the general partial differential equation or partial differential equation [27]. It has strong computing performance and direct coupling ability of multiple physical fields, as well as efficient pre-processing and meshing modules.

In engineering, the cooling system is usually made up of steel pipe with an outer diameter of 28 mm and a wall thickness of 1.6 mm, or PVC pipe with an outer diameter of 28 mm and a wall thickness of 2 mm. For the precise simulation, the first step is to establish the real layout of the cooling system, which can be achieved through internal modeling in COMSOL Multiphysics, or imported from other three-dimensional software.

In COMSOL Multiphysics, Laminar Flow Module and Heat Transfer Module can be directly defined to calculate the water flow in the cooling pipe and heat-flow coupling between cooling water and concrete. Considering that the thickness of the cooling pipe is relatively small, a thin layer is adopted to simulate the cooling pipe. The cooling pipe and the flow boundary share the same set of nodes, which avoids the simplification of the convection boundary and can realistically simulate the influence of the cooling pipe on cooling effects.

Both the Dirichlet and Neumann boundary conditions can be directly defined in the Heat Transfer Module. Furthermore, the direction adjustment of cooling water can be achieved by reversing the water flow and Open boundary condition as follows:(8)$$\begin{array}{ll} {\left.T\right|}_{\Gamma_{1}}=T_{0}\;\mathrm{if}\;n\cdot u\text{<}0 & \\ & \text{(Open boundary condition)}\\ -n\cdot q=0\;\mathrm{if}\;n\cdot u\geq 0 & \end{array}$$

During the calculation, it can automatically transfer the coupling parameters between different modules. Considering that the flow velocity **u** and pressure *p* of cooling water are basically stable, the steady-state solver can be set to solve the fluid field first, and then it can be used to solve the heat-flow coupling field using a transient solver.

## 4. Model Validation and Numerical Analysis

### 4.1. Numerical Verification of Pipe Cooling in Concrete Column

Due to the complexity of heat-flow coupling of pipe cooling in concrete, a theoretical solution cannot be obtained in most cases. In this study, the theoretical solution deduced in Ref. [10] for pipe cooling without considering hydration heat is adopted to verify the precise simulation first.

As shown in Figure 2, metal and PVC pipes (with radius 14 mm and wall thickness 2 mm) are embedded in concrete column (with radius 0.845 m and length 200 m) for cooling, where the water flow is 0.90 m^3^/h. The initial temperature of water and concrete are 4 °C and 20 °C respectively. All the boundaries are supposed to be heat isolated and the calculation time step length is 0.25 days. The basic geometric information and material properties are presented in Figure 2 and Table 1.

The cooling effects of metal and PVC pipes are calculated and demonstrated in Figure 2 respectively. From Figure 2, it can be seen that simulation results are close to those from theoretical solutions in Ref. [10], and the maximum relative error is less than 2%. In addition, it is feasible to simulate the thermal performance of different pipes using a thin layer. It can be concluded that the precise simulation of heat-flow coupling based on COMSOL Multiphysics is correct and reliable.

### 4.2. Comparison of Precise Simulation and Equivalent Algorithm

For the further verification of precise simulation of heat-flow coupling, the following analysis is carried out. As sketched in Figure 3, a concrete block with a length of 32 m, a width of 24 m, and a thickness of 5.6 m are cast in three layers (with thickness 1.8 m, 1.8 m, and 2.0 m) and the cooling system made up with PVC pipe is spatially distributed as 1.0 m in horizontal and consistent with a layer thickness in vertical. Each layer is equipped with three cooling pipes and the length of each pipe is around 245 m. The FE model and representative points and lines are displayed in Figure 3. A precise FE meshing is established, which can represent the actual layout of the cooling system. The casting interval between layers is 21 days and each layer is cooled for 15 days. The inlet water temperature is 10 °C, and the water flow is 1.2 m^3^/h. The thermal properties of rock, concrete, and PVC pipe is shown in Table 1. Numerical simulations are carried out based on both precise simulations and equivalent algorithms. The simulation starts from 1 July and calculation duration is 0~120 days with calculation time step length 0.25 days.

The adiabatic temperature of concrete is expressed in exponential form as following:(9)$$\theta (\tau )=\theta_{0}(1-e^{-mp})$$
where $\theta_{0}=25$ °C is the ultimate adiabatic temperature rise, m=0.35 is a parameter that that describes the dissipation rate of hydration heat, τ is concrete age in days.

The ambient temperature is expressed as:(10)$$T_{tem}=T_{a}+T_{bf}\cos[2\pi /365(t-\tau_{tem})]$$
where $T_{a}=15.6$ °C represents the annual average temperature, $T_{bf}=12.6$ °C is the variation of annual mean temperature, $\tau_{tem}=200$ days indicate the initial phase, t represents time in days since 1 January.

Figure 4 compares the evolution of average temperature calculated from precise simulation and equivalent algorithm. It can be observed that in the early stage, due to the release of hydration heat, the temperature of concrete rises rapidly and subsequently decreases under the effect of pipe cooling and environmental heat dissipation. After 15 days of pipe cooling, the average temperature of concrete rises. This phenomenon can be explained as: the boundary heat dissipation is less than the internal hydration heat release. In addition, the temperature of concrete cast former is affected by those cast later. The heat released from the upper layer conducts to the lower layer, which makes for the temperature increase of the lower layer. On the whole, calculated average temperature from precise simulations and equivalent algorithms show a great coincidence.

Specifically, the temperature evolution of representative points A and B from precise simulations and equivalent algorithms are comparatively illustrated in Figure 5. Representative point A is away from the cooling pipe (located in the middle of cooling pipes) and thus less affected by pipe cooling. In contrast, representative point B lies close to the cooling pipe (only 0.1 m away from the cooling pipe) and is greatly affected by cooling water. Since points A and B are located in the middle of layer 2#, the calculated temperature for points A and B from the equivalent algorithm basically coincide. Nevertheless, the temperature curves of points A and B from the precise simulation are obviously different, where the temperature of point B is significantly lower due to the influence of the pipe cooling.

In order to intuitively describe the temperature gradient inside the concrete, the evolution of temperature distribution along the representative line in layer 2# is illustrated in Figure 6. From Figure 6, it can be seen that precise heat-flow coupling simulation can reflect the temperature gradient around the cooling pipe. The temperature of concrete away from the cooling pipe is significantly higher than that near the cooling pipe and the evolution of temperature gradient is consistent with the evolution of temperature difference between representative points A and B in Figure 5. Furthermore, precise simulation can reflect the temperature rise of cooling water along pipeline: on one hand, the temperature of cooling water increases from 10 °C to about 17 °C from inlet to outlet; on the other hand, the effect of daily adjustment of inlet and outlet can also be modeled. Figure 7 displays the temperature nephogram of concrete after pipe cooling from precise simulation. In conclusion, the precise simulation can provide realistic temperature distribution and evolution around cooling pipes. In engineering practice, the temperature rise of cooling water from inlet to outlet is usually used to evaluate the cooling effect. The internal temperature gradient directly affects the cracking risk and safety of the concrete structure. In general, the temperature difference between cooling water and concrete should be controlled within a certain range.

### 4.3. Sensitivity Analysis of Cooling Schemes

In this section, the cooling effects under different water flow and water temperatures are analyzed and discussed. The calculation model and parameters are the same as those in Section 4.2, only adjusting the water flow (0.60 m^3^/h, 1.20 m^3^/h, and 1.80 m^3^/h) or water temperature (5 °C, 10 °C, and 15 °C). The calculation results of layer 2# during the cooling period from precise simulation under different cooling schemes are displayed in Figure 8 and Figure 9.

It can be seen from Figure 8a that with the increase of water flow, the effect of pipe cooling is strengthened, while the strengthening trend decreases. Figure 8b shows the temperature gradient inside concrete and temperature difference between inlet and outlet under different water flow at 36 days. It can be seen that the change of water flow has little influence on the temperature gradient inside the concrete. Besides, from Figure 8b, it can be seen that the temperature difference between inlet and outlet decreases with the increase of water flow, which indicates that the utilization efficiency of cooling water decreases. In this way, the coupling relationship between heat convection and water flow can be reflected. In this study, water flow at 1.50 m^3^/h can achieve the optimal cooling effect. Therefore, from the perspective of practical engineering, it is rational to control the water flow at around 1.5 m^3^/h.

It can be seen from Figure 9a that the cooling effect by reducing the water temperature is obviously better than increasing the water flow. Combined with the influence of water temperature on the temperature difference between inlet and outlet in Figure 9b, it can be seen that the lower cooling water temperature corresponds to the better cooling effect. However, the lower cooling water temperature may also result in a larger temperature gradient and tensile stress around the cooling pipe, which may induce thermal cracking. Generally speaking, the temperature difference between the cooling water and concrete should not exceed 20 °C.

## 5. Conclusions

Focused on the complex heat-flow coupling problem of pipe cooling in concrete, this paper systematically reviews the development of a calculation method for the thermal field of concrete containing cooling pipe and provides the detailed implementation method of precise simulation. The correctness and feasibility of precise simulation are verified by theoretical solution and equivalent algorithm. Several conclusions can be drawn:(1)In this study, the precise simulation of heat-flow coupling for concrete containing cooling pipe is conducted based on COMSOL Multiphysics, where the influence of pipe materials and water flow direction on cooling effects can be directly modeled.(2)Compared with the equivalent algorithm, the precise simulation can characterize the real layout of the cooling system and the temperature rise of the cooling water along the pipeline, which can be used to evaluate the cooling efficiency. Besides, the precise simulation can reflect the sharp temperature gradient around the cooling pipe, which provides the basis for the subsequent thermal stress analysis.(3)Through the sensitivity analysis of cooling schemes, it can be concluded that in order to improve the utilization efficiency of cooling water, the water flow should be controlled between 1.2 m^3^/s~1.8 m^3^/s. In addition, the temperature of cooling water needs to be controlled to avoid a large temperature gradient.

Nevertheless, there still exist some issues that need to be further explored. The following research will be carried out in follow-up work:(1)Carry out the relevant temperature monitor experiments and make a comparative analysis between monitoring data and simulation results.(2)Extend the model for considering the thermal stress and analyze the influence of different cooling schemes on the cracking risk of mass concrete.

## Figures and Tables

**Figure 1 materials-14-05142-f001:**
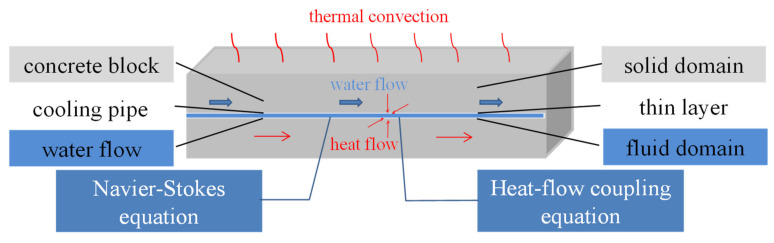
Schematic diagram of heat-flow coupling of pipe cooling in concrete.

**Figure 2 materials-14-05142-f002:**
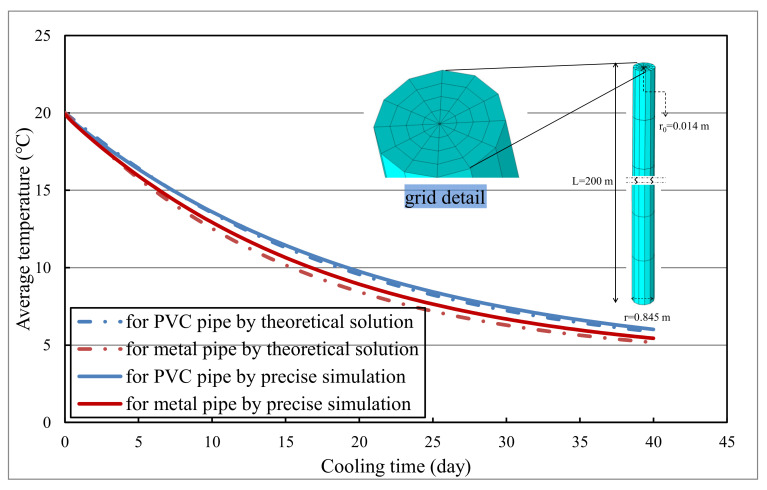
Comparison of pipe cooling from precise simulation and theoretical solution.

**Figure 3 materials-14-05142-f003:**
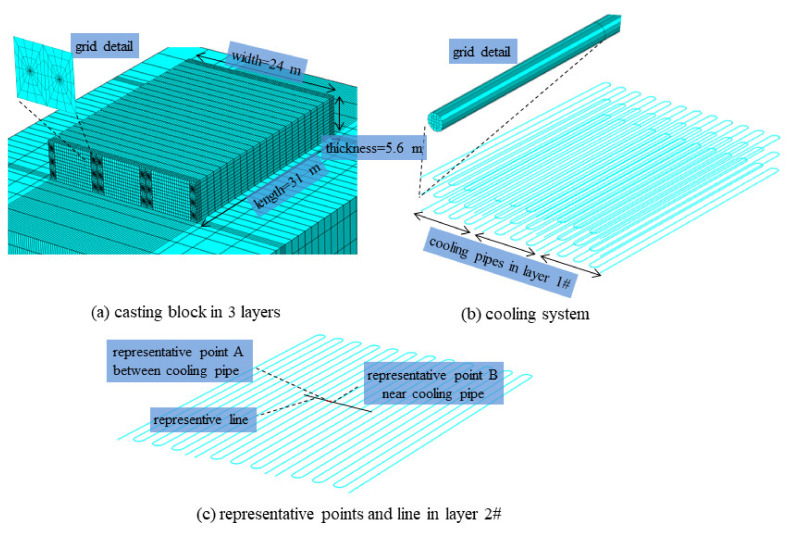
FE model and representative points and line: (**a**) casting block in 3 layers, (**b**) cooling system, (**c**) representative points and line in layer 2#.

**Figure 4 materials-14-05142-f004:**
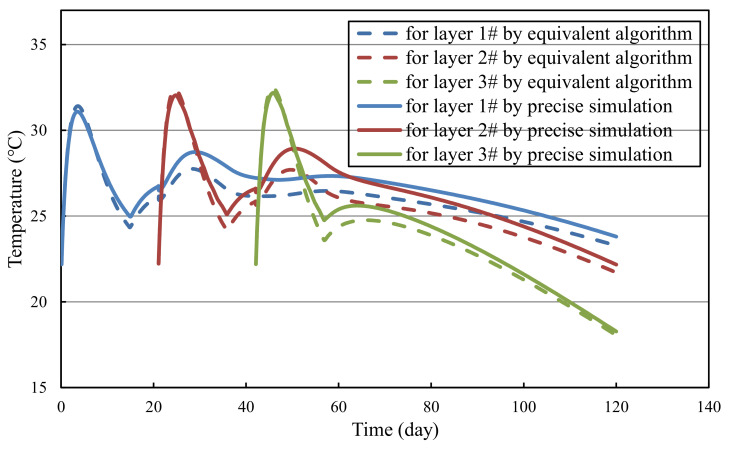
Comparison of calculation results from precise simulation and equivalent algorithm.

**Figure 5 materials-14-05142-f005:**
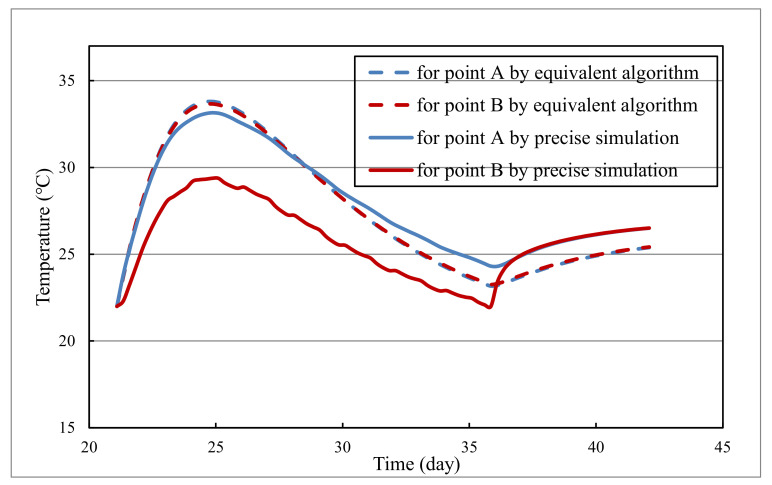
Temperature evolution of representative points A and B.

**Figure 6 materials-14-05142-f006:**
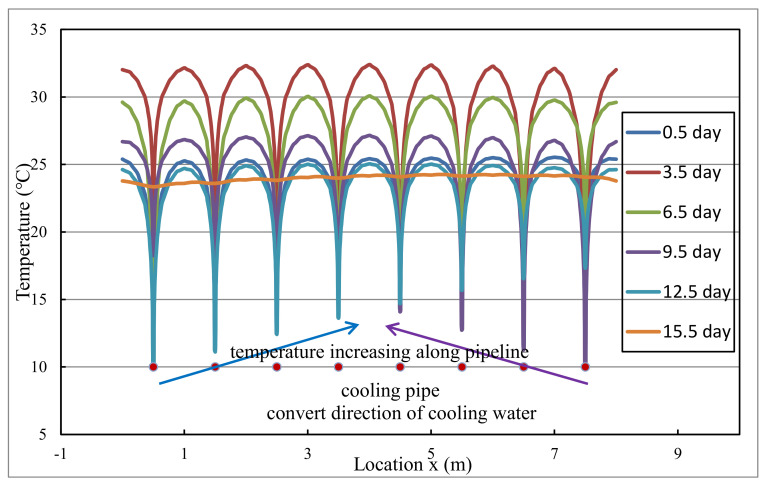
The evolution of temperature distribution along representative line in layer 2#.

**Figure 7 materials-14-05142-f007:**
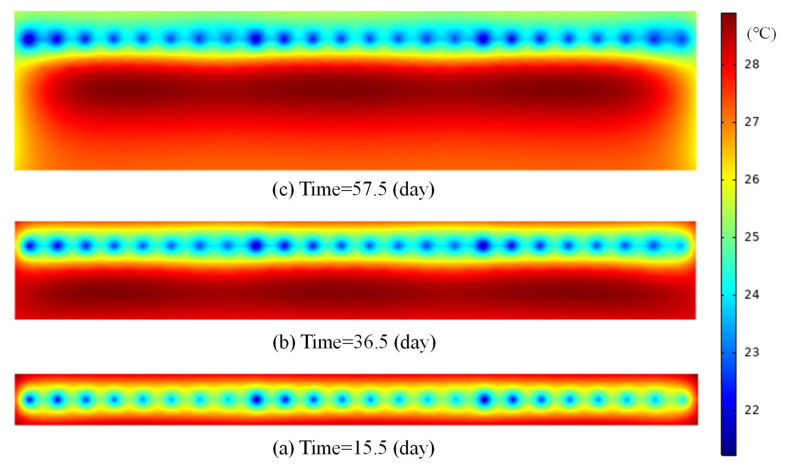
Temperature nephogram of concrete after pipe cooling.

**Figure 8 materials-14-05142-f008:**
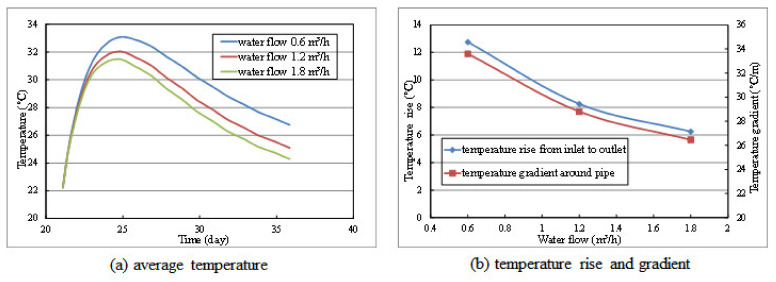
The calculation results of layer 2# under different water flow: (**a**) average temperature, (**b**) temperature rise and gradient.

**Figure 9 materials-14-05142-f009:**
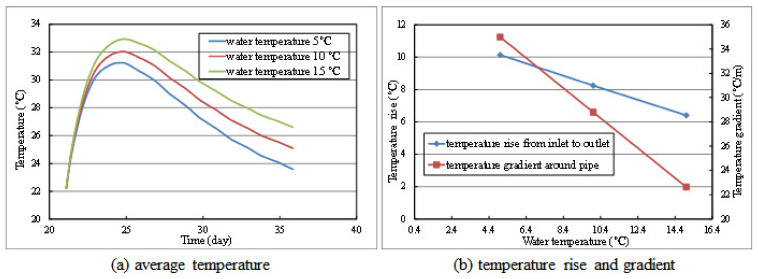
The calculation results of layer 2# under different water temperature: (**a**) average temperature, (**b**) temperature rise and gradient.

**Table 1 materials-14-05142-t001:** Thermal properties of rock, concrete and cooling pipe.

Prosperities	Rock	Concrete	Cooling Water	Pipe	
				Metal	PVC
Thermal conductivity [W/(m·°C)]	2.50	2.33	0.63	40	0.46
Specific heat [J/(kg·°C)]	980	830	4200	4600	1300
Density [kg/m^3^]	2700	2500	1000	6000	1400

## Data Availability

The study did not report any data.

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
