# Peer review of "Precise Simulation of Heat-Flow Coupling of Pipe Cooling in Mass Concrete"

_materials, 2021, doi:10.3390/ma14185142_

Round 1

Reviewer 1 Report

Thanks to the authors for their study on the simulation of heat-flow coupling of pipe cooling in mass concrete. The following comments need to be addressed:

  • In the Abstract, please define what COMSOL stands for? Similar comment for the Introduction.
  • In line 16 and 17 of Abstract please explain what your model for verification was.
  • In the Abstract and last paragraph of Introduction, please address the parameters and variable involved in your study.
  • Please explain how your model results would be affected in case of steel reinforcement presence in concrete.
  • Please point out the limitations of this study at the end of Conclusion and some suggestions for future studies.

Reviewer 2 Report

The article titled "Precise simulation of heat-flow coupling of pipe cooling in mass concrete", with Manuscript ID materials-1310309 is recommended for minor corrections.

According to the authors, For a long time, temperature control and crack prevention of mass concrete is a difficult problem in engineering. For the temperature control and crack prevention, the most effective and common-used method is to embed cooling pipe in mass concrete. At present, there still exist some challenges in precise simulation of pipe cooling in mass concrete, which is a complex heat-flow coupling problem. Numerical simulation is faced with the problem of over-simplification and inaccuracy. In this study, precise simulation of heat-flow coupling of pipe cooling in mass concrete is carried out based on COMSOL Multiphysics. Simulation results are comprehensively verified with results from theoretical solution and equivalent algorithm, which proves the correctness and feasibility of precise simulation. Compared with equivalent algorithm, precise simulation of pipe cooling in mass concrete can characterize the sharp temperature gradient around cooling pipe and the temperature rise of cooling water along pipeline more realistically. In addition, the cooling effects and local temperature gradient under different water flow rate and water temperature are comprehensively studied and related engineering suggestions are given.

The authors are technically sound.

The paper has high significance and well referenced.

An abbreviations list is recommended. Or authors, should double check that all abbreviations are spelt out within the work.

It has good conclusion.

Good relevant equations related to the study is recommended, as only one was presented. More relationships can be correlated. All equations used must be cross-referenced, as used in Section 2.

Good references and good introduction.

Paper has good structure and outline.

Paper presents good research model and key parameters on the study.

Weakness is that the discussion on the analysis does not reflect fully on the results. Can the authors add more results on this in the papers briefly? The results presented are sufficient for this study, but I recommend they add one or two more results. 

The  model should be supported by showing the COMSOL versions  used, It should be clearly stated, and the reference theory documentation cross-referenced.

The study should be validated in a section titled VALIDATION. It could be comparison of some quantities by other researchers with related studies.

Some proof reading and minor English Language check is required.

How does the findings from this study reflect the required fluid flow?

How was the coupling of the pipe cooling conducted? Discuss more on this in the discussion.

Reviewer 3 Report

The submitted paper (Manuscript ID: materials-1310309) titled “Precise simulation of heat-flow coupling of pipe cooling in mass concrete” presents an interesting study about a precise simulation of heat-flow coupling of pipe cooling in mass concrete. The writing style and quality should be improved before publication. Also, the following comments must be addressed in the paper.

  • On line 10, For a long time, temperature control and crack prevention of mass concrete is a difficult “problem” in engineering. Please use a job or exercise instead of a problem as you already mentioned it is difficult.
  • On line 26, the hydration of cement releases a lot of heat, please discard “a lot of” and write the hydration of cement produces heat.
  • Section 3 needs to be improved significantly as it did not present the boundary conditions and explain why the method developed is a precise simulation.
  • On line 144, table 1 should include the Coefficient of linear expansion and Poisson ratio values of the materials.
  • The main drawback of the paper is model is not verified against any practical data. Authors are requested to verify their model with practical data, which can be obtained from existing literature.
  • Point (1) in conclusion is not actually a conclusion, please modify it based on the findings of the study.

Round 2

Reviewer 3 Report

My comments are addressed sufficiently, so, I have no further comment.

Author Response

Thank you for your comments